# Dietary Supplementation of Mixed Organic Acids Improves Growth Performance, Immunity, and Antioxidant Capacity and Maintains the Intestinal Barrier of Ira Rabbits

**DOI:** 10.3390/ani13193140

**Published:** 2023-10-08

**Authors:** Zhixin Lin, Guofeng Yang, Min Zhang, Rui Yang, Yating Wang, Pingting Guo, Jing Zhang, Changkang Wang, Qinghua Liu, Yuyun Gao

**Affiliations:** College of Animal Sciences (College of Bee Science), Fujian Agriculture and Forestry University, Fuzhou 350002, China; zxlin599@163.com (Z.L.); ygf18839143681@163.com (G.Y.); 18305995218@163.com (M.Z.); yr18900298560@163.com (R.Y.); 19859812992@163.com (Y.W.); pingtingguo@fafu.edu.cn (P.G.); jingzhang@fafu.edu.cn (J.Z.); wangchangkangcn@163.com (C.W.)

**Keywords:** Ira rabbit, mixed organic acids, growth performance, immunity, antioxidant, intestinal function

## Abstract

**Simple Summary:**

Diarrhea in weaned rabbits is one of the most common problems in early weaned rabbits, which can lead to slow growth, decreased immunity, and even death. Against the background of a total ban on the use of antibiotics in feed, the development and utilization of antibiotic substitutes have become top priorities. Mixed organic acids (MOAs), as a residue-free green additive, have the ability to enhance immunity and intestinal digestion, regulate pH, and improve intestinal microbiota. Although there are more studies on MOAs, the current domestic and international studies on them mainly focus on pigs and chickens, and the effects on rabbits are less reported. Therefore, this experiment used weaned rabbits as the experimental object to explore the effects of the dietary addition of MOAs on the growth, immunity, and intestinal function of rabbits, to provide a theoretical basis for the application of MOAs in the production of meat rabbits. The results showed that MOAs could improve the growth performance, immunity, and antioxidant ability and maintain the intestinal barrier function of weaned rabbits, which has a promising application prospect in meat rabbit breeding.

**Abstract:**

The aim of this study was to investigate the effects of mixed organic acids (MOAs) on growth performance, immunity, antioxidants, intestinal digestion, and barrier function in Ira rabbits. A total of 192 weaned male Ira rabbits at 35 days of age were randomly assigned to four groups with six replicates of eight rabbits each. The rabbits in the control group (CON) were fed a basal diet, and the antibiotic group (SAL) was fed a basal diet supplemented with 60 mg/kg salinomycin. The test groups were fed a basal diet supplemented with 1000 mg/kg and 2000 mg/kg MOAs (MOA1 and MOA2, respectively). The experiment lasted for 55 days. The results showed that the ADG of Ira rabbits in the SAL group and MOA1 group was higher than that in the CON group (*p* < 0.05). The serum IL-6 and liver MDA levels of Ira rabbits in the SAL group, MOA1 group, and MOA2 group were lower than those in the CON group (*p* < 0.05). In addition, sIgA levels in the jejunal mucosa of Ira rabbits in the SAL group and MOA1 group were increased compared with those in the CON group (*p* < 0.05). Compared with the CON group, the gene expression of IL-6 was decreased (*p* < 0.05) in the jejunal mucosa of Ira rabbits in the SAL, MOA1, and MOA2 groups, while the gene expression of IL-1β tended to decrease (*p* = 0.077) and the IL-10 content tended to increase (*p* = 0.062). Moreover, the gene expression of *ZO-1* in the jejunal mucosa of Ira rabbits was elevated in the MOA1 group compared with the CON group (*p* < 0.05). In conclusion, dietary supplementation with MOAs can improve growth performance, enhance immune function and antioxidant capacity, and maintain the intestinal barrier in weaned Ira rabbits.

## 1. Introduction

With the expanding demand for food variety, the consumption of meat rabbits in recent years has increased greatly. In addition to its food value, rabbit fur can also be used to make leather garments, which shows the promising future of the rabbit farming industry. However, diarrhea is a common problem in the process of meat rabbit breeding, causing significant losses to the rabbit breeding industry. Diarrhea in rabbits is affected by a variety of factors, such as feeding density, environmental temperature and humidity, feed formulation, and weaning [1]. Among them, weaning is the most difficult problem in the process of raising young rabbits. Due to the physiological incompleteness of young rabbits, weak immune function, low resistance, and adaptability to changes in the environment and diet structure, stress reactions such as indigestion, diarrhea, and even death often occur after weaning [2,3]. Although meat rabbits possess a certain degree of resistance during the fattening process and are not as vulnerable as young rabbits, the intestinal health of fattening rabbits should also be emphasized, which is critical for their growth and development [4]. In the past, antibiotics have been used as growth promoters to mitigate the adverse effects of weaning and to maintain intestinal health [5]. However, misuse of antibiotics not only leads to bacterial resistance but also causes residues in animal products and the environment [6]. Therefore, many countries have banned the addition of antibiotic growth promoters (AGPs) to livestock diets [7]. In January 2006, AGPs were banned in the European Union. Since 2020, China has also completely banned the addition of antibiotics to feed. Therefore, there is an urgent need for the exploitation of new feed additives as alternatives to antibiotics in order to maintain the intestinal health of rabbits and ensure their growth and development.

Organic acid, a common additive, is commonly used in piglet feed, poultry feed, silage, and other fields because of its good flavor and strong antibacterial effect [8]. According to different mechanisms of action, organic acids are usually divided into two categories: the first, such as fumaric acid, citric acid, malic acid, lactic acid, and other macromolecular organic acids, can only indirectly reduce the number of harmful pathogens by lowering the pH value of the gastrointestinal environment. This type of organic acid can only play its role in the stomach and cannot lower the pH value in the small intestine because its molecular weight is relatively large; the per unit weight of the acid molecules releases less H^+^, so their pH-lowering effect is also worse than that of small molecules of organic acids [9]. The second type of small-molecule organic acids, including formic acid, acetic acid, and propionic acid, can not only reduce the pH in the environment but also have an inhibitory effect on Gram-negative bacteria because they can damage the bacterial cell membrane to interfere with the synthesis of bacterial enzymes, then affect the replication of the bacterial DNA, and finally produce an anti-Gram-negative bacterial effect [10]. Some scholars tested the antibacterial effects of formic acid, acetic acid, propionic acid, fumaric acid, citric acid, and lactic acid and found that formic acid was the most effective, followed by propionic acid, acetic acid, and fumaric acid [11]. It has been reported that the addition of formic acid to the diet of weaned piglets increased the average daily gain (ADG), decreased the feed-to-weight ratio (F/G), and improved jejunal microbial diversity [12]. Currently, the feeding acidifier in aquaculture production is mainly based on mixed organic acids (MOAs), which overcomes the shortcomings of a single acidifier, such as single function, large additive amount, strong corrosiveness, poor palatability, and greater application value [13]. Many studies have shown that, in addition to reducing gastrointestinal pH, MOAs also have a variety of physiological functions, such as enhancing the body’s immune function, improving the activity of digestive enzymes, and improving the intestinal microbiota. Regarding growth performance, Venkatasubramani et al. [14] found that adding formic acid and propionic acid to the diet increased F/G but ADG and average daily feed intake (ADFI) were not affected in broilers. Unfortunately, most of the studies on the function of MOAs have mostly focused on chickens and pigs, and little research has been reported on their use in rabbit production. According to previous studies, formic acid has a good acidifying effect, which can regulate the balance of intestinal flora [15] and promote the absorption of nutrients in the intestinal tract of animals [16]. Propionic acid can be converted to glucose through the gluconeogenesis pathway, thus promoting the development of the organism [17]. In view of the excellent antimicrobial effect of formic acid and propionic acid, as well as a certain growth-promoting effect, a composite acidifier mixed with formic acid and propionic acid was selected in this experiment, aiming to explore the effects of adding MOAs to diets on the growth performance, immunity, antioxidant activity, intestinal digestion, and barrier function of the Ira rabbit. The effect was also compared with that of salinomycin to provide a theoretical basis for the application of MOAs in rabbit production.

## 2. Materials and Methods

### 2.1. Animals, Experimental Design, and Diet

A total of 192 male rabbits with similar body weight and weaned at the age of 35 ± 2 days were randomly divided into 4 treatments with 6 replicates of 8 rabbits each. The four treatments were as follows: (1) control group (CON, basal diet), (2) antibiotic group (SAL, basal diet + 60 mg/kg salinomycin), (3) MOA1 group (basal diet + 1000 mg/kg MOAs), and (4) MOA2 group (basal diet + 2000 mg/kg MOAs). The basal diet was formulated according to the dietary requirements of China’s agricultural industry standards (NY/T 4049-2021). The diet (Jinhua Long Feed Co., Ltd., Fuzhou, China) was pelleted, and its composition and nutritional levels are shown in Table 1. The diet provided to the rabbits in this study was carefully monitored to ensure that aflatoxin levels were well below the established safety limits for animal feed. This precautionary measure was taken to safeguard the rabbits’ health and welfare. Aflatoxin contamination in animal feed can pose serious health risks, including impaired growth and liver damage [18]. The MOAs (29% formic acid, 6% propionic acid, 30% lignosulfonate, and 35% carrier silica) were provided by Myron (Zhangzhou) Biotechnology Co., Ltd. (Zhangzhou, China). The experiment period was 55 days. The test rabbits were kept in closed rabbit hutches with three layers of cages to ensure normal light, temperature, and ventilation in the hutches. The immunization procedures of the litter were strictly in accordance with the routine method. The specific protocol was as follows: at 25 days of age, 2 mL of polyvalent inactivated *E. coli* vaccine; at 30 days of age, 2 mL of inactivated *bacillus* and *bordetella* vaccine; and at 40 days of age, 2 mL of inactivated rabbit distemper vaccine. All of the above vaccines were administered by subcutaneous injection. The health and mental conditions of the rabbits were observed every day. All rabbits had ad libitum access to feed and water. The indoor temperature was controlled at 24 °C.

### 2.2. Growth Performance

Rabbits from each replication were weighed at 90 days old, and the total feed consumption of each replicate was recorded. The diarrhea of the rabbits was visually checked every day, and records were made. The ADFI, ADG, F/G, and diarrhea rates were calculated.

### 2.3. Sample Collection

On day 90, one rabbit that was close to the average weight of each replicate was selected. Blood was collected from the jugular vein and left at room temperature for 2 h, and serum was separated by centrifugation at 1000× *g* for 15 min at 4 °C. The collected serum samples were stored at −20 °C prior to being tested. After blood collection, rabbits were euthanized, and liver tissue samples were collected and stored at −80 °C for testing antioxidant-related indicators. Tissue was collected from the mid-jejunum (approximately 3 cm), and the segment was then opened longitudinally and gently rinsed with 4 °C phosphate-buffered saline (PBS). A sample of the jejunal mucosa was gently scraped with a sterile slide. The scraped intestinal mucosa was stored in liquid nitrogen and then quickly frozen at −80 °C for further analysis.

### 2.4. Determination of Serum Cytokines

The concentrations of interleukin-6 (IL-6, ml027844), interleukin-10 (IL-10, ml027828), interleukin-1β (IL-1β, ml027836), and tumor necrosis factor-α (TNF-α, ml028087) in serum were measured using ELISA kits purchased from Enzymatic Biotechnology Co., Ltd. (Shanghai, China) according to the manufacturer’s instructions. Serum samples were diluted 5-fold during the assay. Concentrations of the standards in the kit were 10, 20, 40, 80, and 160 pg/mL. More detailed steps can be found in our previous reports [19]. The absorbance was detected by the iMark™ Microplate Absorbance Reader (Bio-Rad, Hercules, CA, USA).

### 2.5. Liver Antioxidant Indicators

Liver tissue was accurately weighed to approximately 0.1 g, and a 10% tissue homogenate was prepared by adding 9 times the volume of saline at a ratio of liver weight (g): volume (mL) = 1:9 and mechanically homogenized (70 Hz, 15 s, 3 times) in an ice-water bath. The resulting homogenate was centrifuged at 1000× *g* for 10 min, the supernatant was collected, and the 10% homogenate was then diluted to 1% with saline. As Kesik et al. [20] reported previously, the total antioxidant capacity (T-AOC, ml094998) and the contents of superoxide dismutase (SOD, ml092620), catalase (CAT, ml095267), glutathione peroxidase (GSH-Px, ml095262), malondialdehyde (MDA, ml094962), glutathione (GSH, ml094991), and oxidized glutathione (GSSG, ml094995) were measured using assay kits from Enzymatic Biotechnology Co., Ltd. (Shanghai, China), and the ratio of GSH/GSSG was calculated to evaluate glutathione redox status. The absorbance was detected by the iMark™ Microplate Absorbance Reader (Bio-Rad, Hercules, CA, USA).

### 2.6. Jejunal Mucosa Enzyme Activity Assay

Referring to Marchioro et al. [21], with some modifications. Lipase (A054-2-1), α-amylase (C016-1-1), and trypsin (A080-2-2) activities of jejunal mucosal samples were assayed using commercial kits (Nanjing Jiancheng Bioengineering Institute, Nanjing, China). Sample pretreatment methods are as follows: A 0.1 g mucosal sample was accurately weighed, 9 times the volume of sample homogenization medium was added, and mechanical homogenization (70 Hz, 15 s, 3 times) was performed to prepare a 10% mucosal homogenate. The mucosal homogenate was centrifuged at 1000× *g* for 10 min, and the supernatant was collected and then assayed for mucosal lipase and trypsin activities as per the instructions. For determination of α-amylase activity, a 0.1 g sample of mucosa was weighed, 1 mL of distilled water was added, and the homogenate was ground. The homogenate was poured into a centrifuge tube, and then the extraction solution was added and left to extract for 15 min at room temperature, with shaking every 5 min to allow full extraction. After extraction, the mixture was centrifuged at 1000× *g* for 10 min at room temperature, the supernatant was collected, distilled water was added to 10 mL, and the mixture was shaken well to obtain the amylase reserve solution. The above amylase stock solution (1 mL) was pipetted, and 4 mL of distilled water was added and shaken well to make an amylase dilution solution. The amylase stock solution and dilution solution were used for the determination of amylase activity. The absorbance was detected by the iMark™ Microplate Absorbance Reader (Bio-Rad, Hercules, CA, USA). The above results were corrected for total protein determination using the BCA protein quantification kit (ZJ101) purchased from Epizyme Biomedical Technology Co., Ltd. (Shanghai, China).

### 2.7. Determination of Jejunal Mucosal Immune Indicators

The jejunal mucosa samples were homogenized using ice-cold saline (mucosa weight (g): saline volume (mL) = 1:9), and then the samples were centrifuged at 1000× *g* for 10 min to collect the supernatant. Total protein was determined using a BCA protein quantification kit (ZJ101) purchased from Epizyme Biomedical Technology Co., Ltd. (Shanghai, China) for correction of subsequent results. The jejunal mucosa immune factor concentrations of IL-6 (ml027844), IL-10 (ml027828), IL-1β (ml027836), and secretory immunoglobulin A (sIgA, ml036798) were measured using ELISA kits purchased from Enzymatic Biotechnology Co., Ltd. (Shanghai, China). The absorbance was detected by the iMark™ Microplate Absorbance Reader (Bio-Rad, Hercules, CA, USA).

### 2.8. qRT‒PCR

The method to determine gene expression was generally the same as we previously reported [22]. Total RNA was isolated from jejunum mucosa using the Trans-Zol UP Plus RNA Extraction Kit (Beijing Quanshijin Biotechnology Co., Ltd., Beijing, China) according to the manufacturer’s instructions. The RNA concentration and purity were assessed using Nanodrop 2000 (Thermo Fisher Scientific Corporation, Wilmington, NC, USA). Subsequently, total RNA was reverse transcribed to cDNA using the PrimeScript RT kit (Promega Biotechnology Co., Ltd., Beijing, China). All primers were designed and synthesized by Fuzhou Shangya Biotechnology Co., Ltd., and the primer sequences are shown in Table 2. The expression levels of jejunal immune factors, tight junction proteins, and the housekeeping gene GAPDH were quantified using SYBR Premix Ex Taq kits (Promega Biotechnology Co., Ltd., Beijing, China) on the CFX ConnectTM Real-Time PCR Detection System (Bio-Rad Laboratories, Inc., Hercules, CA, USA). qRT‒PCR was carried out on the Go Taq^®^ qPCR Master Mix (Promega Biotechnology Ltd., Madison, WI, USA), and the amplification conditions were as follows: initial denaturation at 94 °C for 30 s, followed by 40 cycles of denaturation at 94 °C for 5 s, annealing at 56 °C for 15 s, and extension at 72 °C for 10 s. GADPH was used as a reference gene, and the relative mRNA expression level was calculated using the 2^−∆∆CT^ method.

### 2.9. Data Analysis

Statistical analysis was performed using SPSS, version 26.0 (SPSS, Inc., Chicago, IL, USA). The Shapiro–Wilk test was used to check the normality of all data, and Levene’s test was used to check the homogeneity of variance. A one-way analysis of variance (ANOVA) was used for data analysis, and Tukey’s multiple range tests were used for multiple comparisons. The results are presented as the mean ± standard deviation. *p* < 0.05 was considered statistically significant, and 0.05 ≤ *p* ≤0.10 was taken to indicate a statistical tendency.

## 3. Results

### 3.1. Growth Performance

As shown in Table 3, there was an increasing trend in ADG of Ira rabbits in the MOA group compared to the CON group (*p* = 0.051). However, there was no significant difference in ADFI, F/G, or diarrhea rate of Ira rabbits among groups (*p* > 0.05).

### 3.2. Serum Immunity

Serum IL-6 levels were reduced (*p* < 0.05) in Ira rabbits in the SAL, MOA1, and MOA2 groups compared to the CON group (Figure 1).

### 3.3. Liver Antioxidant Function

As shown in Figure 2, the MDA content in the livers of Ira rabbits in the SAL, MOA1, and MOA2 groups was reduced compared with the CON group (*p* < 0.05). In addition, the content of CAT in the livers of Ira rabbits in the MOA2 group was also higher than that in the CON group (*p* < 0.05).

### 3.4. Digestive Enzyme Activity of Jejunal Mucosa

As can be seen from Figure 3, there was no significant difference in the activity of digestive enzymes in the jejunal mucosa of the Ira rabbits in each group (*p* > 0.05).

### 3.5. The Content of Immune Factors in Jejunal Mucosa and the Expression of Related Genes

As shown in Figure 4, the sIgA content in the jejunal mucosa of Ira rabbit was increased in the SAL and MOA1 groups compared with the CON group (*p* < 0.05), and the IL-10 content in the SAL, MOA1, and MOA2 groups showed an increasing trend (*p* = 0.062). In addition, the gene expression of IL-6 in the jejunal mucosa of Ira rabbits in the SAL, MOA1, and MOA2 groups was lower than that in the CON group (*p* < 0.05), and there was a trend of decreasing gene expression of IL-10 (*p* = 0.077) (Figure 5).

### 3.6. Expression of Tight Junction Protein in Jejunal Mucosa

As shown in Figure 6, the gene expression of *ZO-1* was significantly increased in the jejunal mucosa of Ira rabbits in the MOA1 group compared with the CON group (*p* < 0.05). However, there was no significant change in the gene expression of Occuldin and Claudin-3 (*p* > 0.05).

## 4. Discussion

Organic acids have been extensively studied in livestock, poultry, and aquatic animals. The results of their studies on animal growth performance vary. In an experiment by Kinza Saleem et al. [23], organic acids increased body weight gain and FCR in broilers, while feed intake was not affected. A study on the effect of encapsulated organic acids on piglets showed that encapsulated organic acids enhanced the ADG of piglets. The results of Ma et al. showed that MOAs increased ADG and the final body weight of broilers. Long et al. [24] reported that the addition of MOAs (formic, acetic, and pyruvic acids) to diets increased the ADG of weaned piglets. Similarly, in this study, dietary supplementation with 1000 mg/kg MOAs also increased the ADG of Ira rabbits. In summary, it can be seen that organic acids can improve animal growth performance to some extent, and this positive effect can be attributed to increased nutrient digestibility as well as reduced gastrointestinal pH (inhibiting the survival of pathogenic microorganisms in the gastrointestinal tract that are susceptible to low pH). In contrast, Asriqah et al. [25] showed that the addition of formic, propionic, or butyric acid to the diet of *Clarias gariepinus* had no significant effect on its body weight gain, feed conversion, and survival rate. This may be because, compared to terrestrial animals such as pigs and chickens, aquatic animals are characterized by short and simple digestive organs, poor activity of digestive enzymes, and short retention time of food in the intestinal tract [26]. Organic acids are excreted from the body before they can fully play their role in the digestive tract, so growth performance is not significantly affected by the intake of organic acids.

Antioxidant capacity can reflect the health status of the organism, and the oxidative and antioxidant systems in the organism are jointly involved in regulating the growth and elimination of free radicals [27]. Oxidative stress is one of the common problems that cause great economic losses to animal production, and it is a process of oxidative damage caused by increased generation or decreased scavenging capacity of free radicals in the organism, leading to their accumulation in the body [28]. The antioxidant enzymes GSH-Px, SOD, and CAT are important components of the antioxidant system [29,30,31]. Among them, GSH-Px scavenges lipid hydroperoxides and H_2_O_2_, thereby reducing the damage caused by organic hydroperoxides in the body. In order to inhibit the accumulation of oxygen radicals, SOD catalyzes the conversion of superoxide anion to hydrogen peroxide, which inhibits superoxide anion from causing cellular damage. A large amount of hydrogen peroxide can damage tissue cells, and CAT catalyzes hydrogen peroxide into water and oxygen, which reduces the damage to the organism. T-AOC indicates the total antioxidant capacity of the organism [32], and the MDA content directly determines the degree of lipid peroxidation of the organism, it is an important indicator for the organism to evaluate the damage of lipid oxidation, and also indirectly reflects the degree of damage of the intestinal mucosal cells [33]. In the present experiment, we examined the content of each antioxidant index in the liver of Ira rabbits and found that MOAs increased CAT content and decreased MDA levels. This is similar to the results of Ma et al. [34], which concluded that the addition of MOAs to the diet increased the levels of T-AOC, SOD, and CAT in the serum of broilers, which in turn enhanced the antioxidant capacity of broilers. In addition, it was also reported that the addition of MOAs to the diet increased serum T-AOC and decreased serum H_2_O_2_ concentration, thereby alleviating oxidative stress in broilers [35]. Therefore, MOAs may enhance the antioxidant capacity of livestock and poultry by increasing the level of antioxidant enzymes. However, there are fewer studies on the specific regimes of MOAs to enhance antioxidant capacity, but it has been demonstrated that propionic acid (one of the components of the MOAs used in this experiment) promotes the nuclear translocation of Nrf2, which activates the antioxidant pathway Keap1/Nrf2 and thus increases the synthesis of downstream antioxidant enzymes [36].

The intestinal tract is the main place for digestion and absorption of nutrients in rabbits and an important gateway to resist the invasion of foreign pathogenic microorganisms [37]. Therefore, ensuring the health of the intestinal tract is of great significance to the production of rabbits. Therefore, this experiment explored the effects of MOAs on the immune function, digestive ability, and barrier function of the intestinal tract of rabbits to evaluate the health effects of MOAs on the intestinal tract.

Intestinal digestive enzyme activity is an important indicator of intestinal digestive capacity [38]. Lipase, α-amylase, and trypsin are the main enzymes for digesting fats, starch, and proteins, and their activities can indirectly reflect the degree of digestion and absorption of nutrients by the organism [39]. These three enzymes perform different functions in the intestine. For example, intestinal lipase, mainly pancreatic lipase secreted by the pancreas, breaks down triglycerides into glycerol and fatty acids, thereby providing energy [40]. α-Amylase breaks down starch into glucose and maltose, thereby providing energy and promoting growth [41]. Trypsin breaks down proteins into amino acids and peptides, which are then absorbed from the intestines into various tissues of the body [42]. The activity of these enzymes is reduced at a higher gastrointestinal pH [43]. Organic acid, when introduced into the organism, can provide exogenous hydrogen ions, increase the acidity of the surimi, create a suitable pH environment for the digestion of digestive enzymes, increase enzyme activity, and enhance the digestibility of nutrients [44]. Ma et al. [34] found that amylase activity in the pancreas of broiler chickens was significantly increased by the addition of 3000 mg/kg MOAs to their diets. Another study reported that the addition of MOAs to the diet increased the activities of trypsin, lipase, and amylase in the pancreas of piglets but had no effect on the activities of digestive enzymes in the jejunum and ileum [45]. However, in the present experiment, the dietary addition of MOA had no significant effect on jejunal digestive enzyme activities in Ira rabbits. This may be because mixed organic acids act on the stomach and foregut after ingestion and do not fully reach and regulate the activity of digestive enzymes in the hindgut. Therefore, the use of techniques such as encapsulation may allow the mixed organic acids to act consistently and stably throughout the intestinal tract of broilers.

The mortality rate of rabbits caused by diarrhea is extremely high. Since the banning of antibiotics in feed, the mortality rate of diarrhea caused by bacterial enteritis in rabbits accounts for the main factor, so the important guarantee to improve the survival rate of rabbits is to strengthen the prevention of bacterial enteritis in rabbits. Bacterial enteritis is due to microorganisms or pathogens invading the intestine, stimulating immune cells, and leading to the production of a large number of inflammatory cytokines and mediators. Inflammatory molecules will destroy the tight junctions, exacerbating the damage to the intestinal mucosal barrier function [46]. Therefore, the intestinal immune barrier and mechanical barrier are crucial in the resistance of rabbits to bacterial enteritis.

The intestinal mucosal immune barrier is mainly composed of intestinal-associated lymphocyte tissues and diffuse immune cells [47]. When antigens such as bacteria, viruses, and toxins stimulate the intestinal mucosa, the intestinal mucosal immune system produces a series of immune substances such as immunoglobulins and cytokines for the effective removal of antigens [48]. In our study, the addition of 1000 mg/kg MOAs to the diet increased the sIgA content in the jejunal mucosa and decreased the mRNA expression of IL-6. sIgA is produced in the intestinal lamina propria and processed by intestinal epithelial cells before being secreted into the intestinal lumen, blocking the adhesion of pathogens or toxins to the mucosa, and it plays a key role in intestinal immunity [49]. IL-6, also known as the B-cell differentiation factor, is an important pro-inflammatory cytokine whose dysregulated expression can cause many diseases or lead to their deterioration [50]. Tong et al. [51] found that propionic acid could inhibit the mRNA levels of IL-6, IL-1β, and TNF-α and slow down the development of intestinal cancers in mice. Wang et al. [52] reported that the feeding of organic acids could reduce the expression of TNF-α in the jejunum of weaned piglets. Wang et al. reported that feeding organic acids reduced the expression level of TNF-α in the jejunum of weaned piglets. Therefore, MOA can promote the secretion of anti-inflammatory cytokines and immunoglobulins in the intestinal mucosa and reduce the gene expression of pro-inflammatory cytokines, thus enhancing jejunal immunity and improving the intestinal immune function of Ira rabbits.

The intestinal mechanical barrier is composed of intestinal mucosal epithelial cells and intercellular tight junctions [53]. Intercellular junctions include tight junctions, gap junctions, adhesion junctions, and bridging junctions [54]. Among them, tight junctions are widely present at the very top of epithelial or endothelial cell junctions and consist of various proteins such as Occludin, Claudin, ligand adhesion molecules, and ZO, which play an important role in intestinal barrier function. Tong et al. [51] found that propionic acid significantly increased the gene expression of ZO-1 and Occludin, improthe intestinal barrier function, and alleviated intestinal inflammation in mice. Diao et al. [55] found that gastric infusion of short-chain fatty acids (acetic, propionic, and butyric acids) increased the gene expression of Occludin and Claudin-1 genes in the duodenum and ileum and promoted the intestinal development of weaned piglets. The results of this experiment showed that the addition of MOA to the diet increased the gene expression of ZO-1 in the jejunal mucosa, which improved the mechanical barrier function of the jejunum, enhanced the intestinal resistance to pathogenic microorganisms, and ensured the balance of the intestinal environment.

It can be seen that MOAs maintain both immune and mechanical barriers in rabbits, which lays a theoretical foundation for their use in rabbits against bacterial enteritis in breeding production. The protective effect of MOAs on the barrier is closely related to their bacteriostatic ability. Studies have shown that MOAs can lower the pH of the gastrointestinal tract, which not only helps to stimulate the metabolic activity of beneficial bacteria such as lactobacilli, accelerating their growth and reproduction but also inhibits the growth and reproduction of bacteria with neutral or alkaline internal environments (most of them are pathogenic bacteria) [15,56]. However, the specific mechanism of MOAs in maintaining the intestinal barrier needs to be further investigated.

## 5. Conclusions

In conclusion, dietary supplementation with MOA reduced serum IL-6 levels and MOA content in the liver and enhanced immunity and antioxidant capacity. In addition, MOA could increase the secretion of sIgA, inhibit the gene expression of IL-6, and promote the gene expression of ZO-1 in the jejunal mucosa, thus maintaining the integrity of the intestinal immune system and mechanical barriers in rabbits. The antioxidant, immune, and intestinal barrier-maintaining effects of MOAs collectively improved the growth performance of Ira rabbits.

## Figures and Tables

**Figure 1 animals-13-03140-f001:**
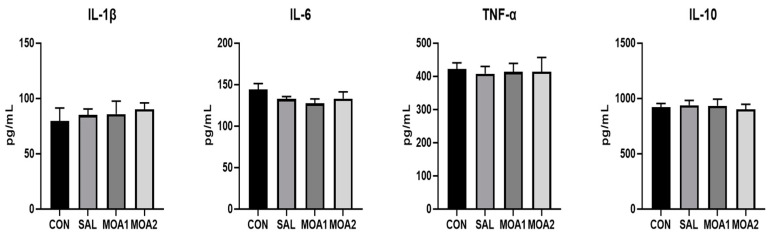
Effects of MOAs on serum immune indices of Ira rabbits. Control group (CON, basal diet); Antibiotic group (SAL, basal diet + 60 mg/kg salinomycin); MOA1 group (basal diet + 1000 mg/kg MOAs); MOA2 group (basal diet + 2000 mg/kg MOAs). Values are the mean ± SD, n = 6.

**Figure 2 animals-13-03140-f002:**
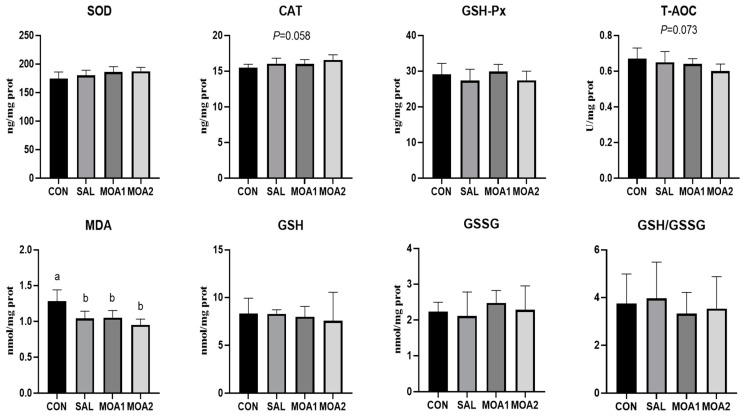
Effects of MOAs on liver antioxidant function of Ira rabbits. Control group (CON, basal diet); Antibiotic group (SAL, basal diet + 60 mg/kg salinomycin); MOA1 group (basal diet + 1000 mg/kg MOAs); MOA2 group (basal diet + 2000 mg/kg MOAs). Values are the mean ± SD, n = 6. The different superscript small letters were judged as a significant difference, *p* < 0.05.

**Figure 3 animals-13-03140-f003:**
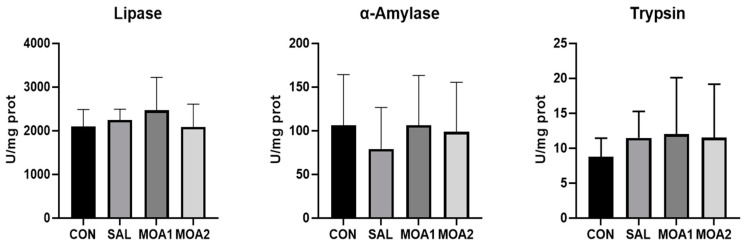
Effects of MOAs on jejunal mucosa digestive enzyme activity in Ira rabbits. Control group (CON, basal diet); Antibiotic group (SAL, basal diet + 60 mg/kg salinomycin); MOA1 group (basal diet + 1000 mg/kg MOAs); MOA2 group (basal diet + 2000 mg/kg MOAs). Values are the mean ± SD, n = 6.

**Figure 4 animals-13-03140-f004:**
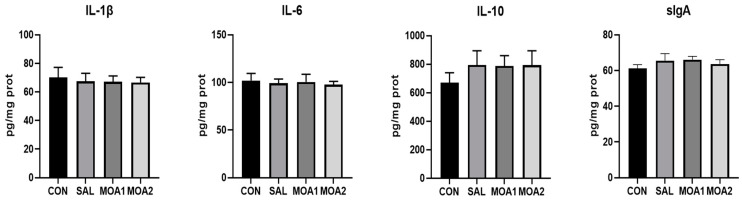
Effects of MOAs on cytokines and sIgA in the jejunum mucosa of Ira rabbits. Control group (CON, basal diet); Antibiotic group (SAL, basal diet + 60 mg/kg salinomycin); MOA1 group (basal diet + 1000 mg/kg MOAs); MOA2 group (basal diet + 2000 mg/kg MOAs). Values are the mean ± SD, n = 6.

**Figure 5 animals-13-03140-f005:**
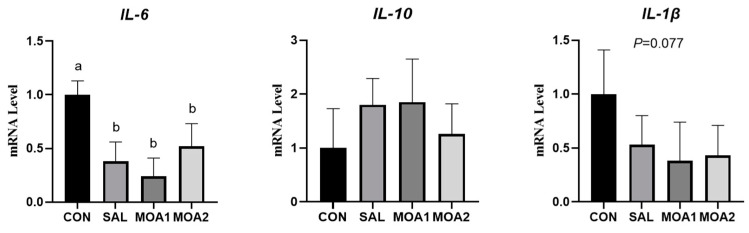
Effect of MOAs on cytokine mRNA expression in the jejunum mucosa of Ira rabbits. Control group (CON, basal diet); Antibiotic group (SAL, basal diet + 60 mg/kg salinomycin); MOA1 group (basal diet + 1000 mg/kg MOAs); MOA2 group (basal diet + 2000 mg/kg MOAs). Values are the mean ± SD, n = 6. The different superscript small letters were judged as a significant difference, *p* < 0.05.

**Figure 6 animals-13-03140-f006:**
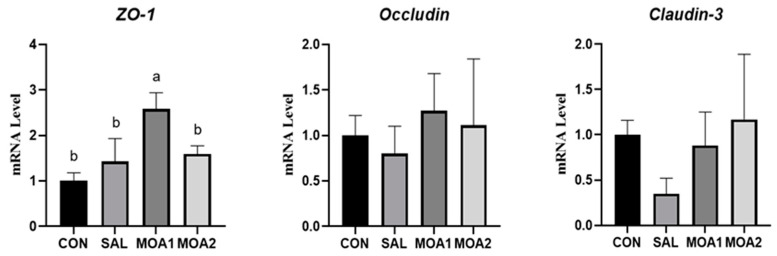
Effects of MOAs on tight junction protein mRNA expression in the jejunum mucosa of Ira rabbits. Control group (CON, basal diet); Antibiotic group (SAL, basal diet + 60 mg/kg salinomycin); MOA1 group (basal diet + 1000 mg/kg MOAs); MOA2 group (basal diet + 2000 mg/kg MOAs). Values are the mean ± SD, n = 6. The different superscript small letters were judged as a significant difference, *p* < 0.05.

**Table 1 animals-13-03140-t001:** Composition and nutrient levels of basal diets (air-dry basis, %).

Items	Content
Ingredients	
Alfalfa meal	38.00
Corn	9.00
Wheat bran	16.90
Wheat DDGS	6.00
Rice husk powder	9.00
Soybean meal	5.00
Rice bran meal	10.00
Wheat middling	3.00
Limestone	1.20
Methionine	0.10
Lysine	0.20
NaCl	0.60
Premix ^1^	1.00
Total	100.00
Nutrient levels ^2^	
Digestible energy (MJ/kg)	9.96
Crude protein	15.48
Crude Fiber	17.35
Neutral detergent fiber	33.56
Acid detergent fiber	20.51
Acid detergent lignin	5.90
Calcium	0.90
Total Phosphorus	0.51
Lysine	0.74
Methionine + Cystine	0.51

^1^ The Premix Provided the following per kilogram of diet: vitamin A, 12,000 IU; vitamin D3, 900 IU; vitamin E, 50 mg; vitamin K3, 1 mg; vitamin B1, 1 mg; vitamin B2, 3 mg; vitamin B6, 1 mg; vitamin B12, 0.01 mg; nicotinic acid, 30 mg; pantothenic acid, 8 mg; folic acid, 0.2 mg; biotin, 0.08 mg; choline chloride, 100 mg; copper, 10 mg; ferrous, 50 mg; manganese, 8 mg; zinc, 50 mg; iodide, 1 mg; selenium, 0.05 mg; cobalt, 0.25 mg. ^2^ DE and nutrients were calculated values.

**Table 2 animals-13-03140-t002:** Nucleotide sequences of primers for quantitative real-time PCR assay.

Gene	Primer Sequence (5′ → 3′)	GeneBank
*GADPH*	F: 5′–GGTAGTGAAGGCTGCTGCTGATG–3′	NC_003074.8
	R: 5′–GTCTCGCACTCCAATCTCTGTTCC–3′	
*IL-6*	F: 5′–ACGATCCACTTCATCCTGCG–3′	NM_001082064.2
	R: 5′–GGATGGTGTGTTCTGACCGT–3′	
*IL-10*	F: 5′–TCACCGATTTTCTCCCCTGTG–3′	XM_051820557.1
	R: 5′–ATGTCAAACTCATGGCTT–3′	
*IL-1β*	F: 5′–TCTGCAACACCTGGGATGAC–3′	XM_051828526.1
	R: 5′–TCAGCTCATACGTGCCAGAC–3′	
*Occludin*	F: 5′–CCGTATCCAGAGAGTCCTACAAGT–3′	XM_008262318.3
	R: 5′–GTCCGTCTCGTAGTGGTCTT–3′	
*ZO-1*	F: 5′–CCGCTCATACCTTCCTCTCA–3′	XM_051822268.1
	R: 5′–GTCATTCACCTCCTTCTTGTTCTC–3′	
*Claudin-3*	F: 5′-CCATCATCCAGGACTTCTACAAC–3′	XM_002721962.4
	R: 5′-AGTAGGCGATCTTGGTGGTC–3′	

**Table 3 animals-13-03140-t003:** Effects of MOAs on the growth performance of Ira rabbits.

Items ^2^	Groups ^1^				*p*-Value
	NC	PC	MOA1	MOA2	
IBW/g	698.25 ± 4.95	696.50 ± 3.45	700.21 ± 4.75	699.31 ± 2.64	0.446
FBW/g	2272.11 ± 107.51	2415.58 ± 137.76	2454.57 ± 109.50	2372.22 ± 94.01	0.061
ADFI/g	114.74 ± 7.35	116.21 ± 8.27	119.39 ± 3.45	117.07 ± 5.53	0.671
ADG/g	28.48 ± 2.41	31.23 ± 2.89	32.13 ± 2.34	29.73 ± 1.56	0.065
F/G	3.76 ± 0.22	3.73 ± 0.22	3.73 ± 0.26	3.94 ± 0.14	0.772
Diarrhea rate/%	4.19 ± 2.83	2.03 ± 1.16	2.00 ± 0.79	2.81 ± 1.29	0.124

IBW = initial body weight; FBW = final body weight; ADFI = average daily feed intake; ADG = average daily gain; F/G: feed-to-gain ratio.,^1^ Control group (CON, basal diet); Antibiotic group (SAL, basal diet + 60 mg/kg salinomycin); MOA1 group (basal diet + 1000 mg/kg MOAs); MOA2 group (basal diet + 2000 mg/kg MOAs). ^2^ Values are the mean ± SD, n = 6.

## Data Availability

The datasets used and analyzed during the current study are available from the corresponding author on reasonable request.

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
