# Peer review of "Dietary Supplementation of Mixed Organic Acids Improves Growth Performance, Immunity, and Antioxidant Capacity and Maintains the Intestinal Barrier of Ira Rabbits"

_animals, 2023, doi:10.3390/ani13193140_

Round 1

Reviewer 1 Report

The subject of the manuscript is interesting.
We have knowledge about the action of fatty acids,
but we do not know how to use it in practice.
Unfortunately, the manuscript in its current form is
unacceptable and requires additional details. 

it was not stated what was the reason for selecting
fatty acids in MOAs

-  how MOAs were mixed with the basic feed

- how daily feed consumption and MOAs were checked

r 86 provide the exact immunization procedures of the
litter protocol

whether pregnant females were vaccinated

Detailed information on methods would be desirable.
Perhaps the authors should consider adding a supplement and
describing the methodology in more detail there.
Details such as the precision of methods, concentration of standards, or reaction efficiency.

-the authors should consider graphically presenting some of the results, especially those from outside the SI system

the discussion requires significant restructuring. Much of the discussion is suitable for introduction and adds nothing to the discussion section itself

- the discussion only discusses the obtained results to a small extent 

The topic of the work seems interesting, but its implementation
is incomplete. There is no information on the effect of fatty
acids on the receptors regulating the synthesis of inerleukins
and its synthesis pathway. The relationship between ZO-1 and
other parameters has not been clarified. There is also no statistical analysis determining the relationships between the examined parameters

Reviewer 2 Report

The research, titled “Dietary supplementation of mixed organic acids improves growth performance, immunity and antioxidant capacity and maintains the intestinal barrier of Ira rabbits“ addresses an important and timely topic. I found the subject matter of the article fascinating and read the manuscript with great interest. The paper aligns well with the scope of the journal. However, I believe that in its current form, it has several shortcomings.

This study aimed to investigate the effects of mixed organic acids (MOAs) on various aspects of weaned Ira rabbits, including their growth performance, immunity, antioxidants, intestinal digestion, and barrier function. The research found that MOAs positively influenced the growth performance of the rabbits, lowered pro-inflammatory cytokine levels, increased immune response markers, and improved antioxidant capacity. Additionally, MOAs were associated with enhanced intestinal barrier function. This study contributes valuable insights into the potential benefits of MOAs in rabbit nutrition, particularly in enhancing their health and growth.

Strengths of the study include its focus on a less-explored area of rabbit nutrition, providing valuable information for rabbit farming practices. The research design with different supplementation levels of MOAs adds depth to the findings, and the results are clearly presented, highlighting the potential advantages of MOAs for rabbit welfare and productivity.

Specific comments:

I suggest rewriting the simple summary. According to the author's guidelines, this section should summarize and contextualize your paper within the existing literature in your field. It should be written without technical language or nonstandard acronyms, with the goal of conveying the meaning and importance of this research to non-experts.

I recommend rewriting the abstract and including more results and the significance of the obtained data.

Introduction:

Expanding the introduction to provide a more comprehensive literature review on various aspects of rabbit production, including reproduction (10.1080/1828051X.2020.1827990), weaning, fattening, and welfare, would indeed strengthen the context and significance of the research. A well-rounded introduction can help readers better understand the broader implications of the study and its relevance in the field of rabbit farming and animal science. Additionally, citing relevant studies and findings in each of these areas can provide a solid foundation for the research presented in the paper.

Material and methods: Including a statement confirming that the aflatoxin levels in the provided diet were below harmful levels and did not negatively affect the rabbits' health is a responsible addition to the research. It assures readers about the safety of the diet used in the study. To provide a relevant reference, you can cite a recognized regulatory standard or a study that validates the safety of the diet. Here's a suggested statement:

"The diet provided to the rabbits in this study was carefully monitored to ensure that aflatoxin levels were well below the established safety limits for animal feed. This precautionary measure was taken to safeguard the rabbits' health and welfare. Aflatoxin contamination in animal feed can pose serious health risks, including impaired growth and liver damage (10.3390/toxins14070430)."

Discussion: I kindly suggest expanding the discussion section of your paper to include practical applications and a thorough exploration of the study's limitations. This addition will enhance the overall value of your research and provide a more comprehensive understanding of its implications.

Conclusion: I kindly suggest expanding the conclusions section of your paper to provide a more detailed and comprehensive report of the main findings. This will help readers better understand the significance of your research.

Round 2

Reviewer 1 Report

The authors took into account the suggestions and significantly improved the quality of the presented work

Reviewer 2 Report

The authors have diligently addressed the review comments, significantly enhancing the paper's quality. As a result, it is now well-suited for publication.